# Chemerin Isoforms and Activity in Obesity

**DOI:** 10.3390/ijms20051128

**Published:** 2019-03-05

**Authors:** Christa Buechler, Susanne Feder, Elisabeth M. Haberl, Charalampos Aslanidis

**Affiliations:** 1Department of Internal Medicine I, Regensburg University Hospital, 93053 Regensburg, Germany; susanne.feder@klinik.uni-regensburg.de (S.F.); haberl.elisabeth@gmx.de (E.M.H.); 2Institute of Clinical Chemistry and Laboratory Medicine, Regensburg University Hospital, 93053 Regensburg, Germany; charalampos.aslanidis@klinik.uni-regensburg.de

**Keywords:** proteolysis, Tango bioassay, biologic activity, chemerin receptors

## Abstract

Overweight and adiposity are risk factors for several diseases, like type 2 diabetes and cancer. White adipose tissue is a major source for adipokines, comprising a diverse group of proteins exerting various functions. Chemerin is one of these proteins whose systemic levels are increased in obesity. Chemerin is involved in different physiological and pathophysiological processes and it regulates adipogenesis, insulin sensitivity, and immune response, suggesting a vital role in metabolic health. The majority of serum chemerin is biologically inert. Different proteases are involved in the C-terminal processing of chemerin and generate diverse isoforms that vary in their activity. Distribution of chemerin variants was analyzed in adipose tissues and plasma of lean and obese humans and mice. The Tango bioassay, which is suitable to monitor the activation of the beta-arrestin 2 pathway, was used to determine the ex-vivo activation of chemerin receptors by systemic chemerin. Further, the expression of the chemerin receptors was analyzed in adipose tissue, liver, and skeletal muscle. Present investigations assume that increased systemic chemerin in human obesity is not accompanied by higher biologic activity. More research is needed to fully understand the pathways that control chemerin processing and chemerin signaling.

## 1. Introduction

The protein chemerin is a chemoattractant for immune cells and it plays a role in adaptive and innate immunity [1,2]. Chemerin is also an adipokine that regulates angiogenesis, adipogenesis, and energy metabolism, which demonstrates a multifaceted function of this protein [3,4,5,6] (Figure 1). Positive correlations of systemic chemerin with obesity related phenotypes, such as insulin resistance, body mass index (BMI), and serum triglycerides, suggest a function of this adipokine in metabolic diseases [2]. Chemerin deficient mice had higher hepatic gluconeogenesis and increased skeletal muscle glucose uptake. In the null mice, the phosphorylation of protein kinase B (Akt) was improved in the muscle upon insulin injection. Of note, glucose stimulated insulin release of pancreatic beta-cells was impaired in the knock-out animals. Fat pad weight was not changed in the null mice, and serum leptin and adiponectin levels were also normal. Interestingly, there were less detectable adipose tissue macrophages. Although this suggests improved insulin sensitivity, insulin induced Akt phosphorylation was reduced in the fat tissue [7]. A separate study describes that the injection of recombinant chemerin reduced serum insulin and tissue glucose uptake in the obese mice but had no effect in the normal-weight animals [8]. In low density lipoprotein (LDL) receptor deficient mice, the overexpression of chemerin was found to induce insulin resistance in muscle, but not the liver or gonadal fat. There were no changes in body weight, levels of serum lipids, and severity of atherosclerosis [9].

Chemerin-stimulated angiogenesis was illustrated in-vitro and in-vivo [10]. Enhanced angiogenesis and increased endothelial-monocyte adhesion upon chemerin incubation indicate a proatherogenic role of this adipokine [11] (Figure 1).

Data on the role of chemerin in metabolic disease are not conclusive so far (Figure 1). Chemerin most likely impairs skeletal muscle insulin sensitivity, although it seems to have a modulatory role in the liver and adipose tissues. The overexpression of chemerin was shown to increase glucose induced insulin secretion, whereas the injection of recombinant protein blocked this process in the pancreatic beta-cell of mice [7,8].

Duration of chemerin signaling, the concentration of chemerin, cell type/tissue analyzed, chemerin processing, and chemerin receptor expression may vary in the different experiments. Pathological characteristics of the murine models used may further modify chemerin signaling [2].

The G protein-coupled receptor chemokine-like receptor 1 (CMKLR1) is one of the two described chemerin receptors with signaling activity so far. The second one is G protein-coupled receptor 1 (GPR1) [6,12,13]. Chemokine receptor-like 2 (CCRL2) is an atypical chemokine receptor that does most likely not exert any signaling activities [6,12,13]. CCRL2 is supposed to present chemerin to CMKLR1 and possibly to GPR1 [14]. Chemerin binds with low nanomolar affinity to all of these receptors [15]. The binding of chemerin to CMKLR1 activated the three Gαi subtypes and the two Gαo isoforms. Chemerin stimulated the rise of intracellular calcium and the decline of cyclic AMP in CMKLR1 expressing CHO cells that are dependent on Gαi signaling [13]. None of the G-proteins were activated upon binding of chemerin to GPR1 or CCRL2 [15]. Nevertheless, the recruitment of beta-arrestin 1 and 2 was observed for both signaling competent receptors [15]. Chemerin further uses the RhoA and Rho-associated protein kinase-dependent pathway downstream of GPR1 and CMKLR1 to activate the transcriptional regulator serum-response factor [16].

Extracellular regulated kinase (ERK) 1/2 was phosphorylated upon chemerin treatment in various cells, including endothelial cells, adipocytes, and skeletal muscle cells. The activation of ERK1/2 at low, but not high chemerin concentration, was described in adipocytes [4,17,18]. Chemerin further activated p38 mitogen-activated protein kinase, Akt and phosphoinositide 3-kinase [2,19]. Short-time incubation with chemerin was shown to activate Akt in hepatocytes, whereas prolonged treatment up to two hours led to a decline of phosphorylated Akt in these cells [20]. Notably, one study reports that chemerin binding weakened the association of phosphatase and tensin homolog (PTEN) with CMKLR1. This enhanced the activity of PTEN and subsequently led to decreased Akt phosphorylation [20]. The activation of the protein kinase C by chemerin stimulated the internalization of CMKLR1. Blockage of this pathway enhanced calcium flux and ERK phosphorylation, showing that this mechanism terminates signaling via desensitization [21]. The nuclear factor kappa B (NFkB) pathway is activated by chemerin in skeletal muscles cells [18]. In adipocytes, the inhibition of chemerin signaling increased NFkB activity [22]. Chemerin thus activates various signaling pathways and the effects depend on incubation time and dose.

As an adipokine chemerin is released by adipocytes [23], but also hepatocytes produce considerable levels of the protein [24]. Serum chemerin is increased in overweight/obesity and correlations with obesity associated traits, like low grade inflammation, blood pressure, and insulin resistance, were identified in some but not all of the patient cohorts studied [1,25,26,27,28,29,30]. Hence, the associations of systemic chemerin levels with the metabolic syndrome are not fully resolved [1,25,26,27,28,29,30]. The serum chemerin levels are heritable with about 16% to 25% of variations being attributed to genetic factors. Polymorphisms in the gene encoding chemerin (retinoic acid receptor responder 2, *RARRES2*) were linked to increased systemic chemerin levels, visceral fat mass, and a higher incidence of the metabolic syndrome [31,32,33,34].

Positive correlations of systemic chemerin with inflammatory cytokines and C-reactive protein were described in chronic inflammatory diseases [35,36,37]. Chemerin has a role in the pathophysiology of rheumatoid arthritis, inflammatory bowel disease, psoriasis, and chronic renal disease [35,36,37]. More recent findings indicate a function of chemerin in cancer, and the pro- as well as anti-carcinogenic effects have been identified [38]. It was shown that chemerin suppressed hepatocellular carcinoma growth but enhanced squamous cell carcinoma migration [39,40].

Whether chemerin is a pro- or anti-inflammatory protein is still debated. Discordant results were obtained in cell culture studies and animal models [5,12]. Murine (m) Chem156, which is a highly active chemerin isoform, antagonized the activation of peritoneal macrophages that were triggered by lipopolysaccharide (LPS) or interferon gamma [41]. Anti-inflammatory effects of mChem156 were also described in a mouse model of acute lung inflammation induced by LPS [42]. In a colitis model, mChem156 aggravated inflammation and suppressed M2 polarization of macrophages [43].

Chemerin is secreted as an inactive precursor and it is activated through C-terminal processing by proteases. Thereby, different isoforms are generated, which have varying biological effects [5,12]. Chemerin derived C-terminal peptide chemerin 15 acted via CMKLR1 and suppressed inflammation [41]. The effects were already obvious at picomolar concentrations of the peptide demonstrating its potent activity [41].

This review article briefly describes the various C-terminal processing forms of chemerin. The expression of chemerin and its receptors, the distribution of chemerin isoforms, and analysis of chemerin bioactivity in obesity will be addressed in detail.

## 2. Expression of Chemerin and its Receptors in Different Tissues

### 2.1. Chemerin Expression in Adipose Tissues

Chemerin is most abundantly expressed in white adipose tissues, in the liver, and to a lesser extent in brown adipose tissue, the lung, skeletal muscles, the kidney, ovary, and the heart [2,25]. Serum chemerin levels are increased in obesity as a result of enhanced synthesis in fat tissues and possibly the liver [2]. The explants of adipose tissues from obese donors indeed released higher chemerin protein amounts than fat tissues that were obtained from lean individuals [18].

#### 2.1.1. Expression of Chemerin in Human Tissues

Chemerin mRNA expression in subcutaneous and omental adipose tissues of patients was higher in obesity, and it decreased upon bariatric surgery evoked weight loss. Serum chemerin levels changed accordingly, indicating that adipose tissue is the main site controlling circulating protein levels in obesity [44]. The inflammatory factors tumor necrosis factor (TNF) and LPS enhanced chemerin production in adipocyte, thus demonstrating a close association between adipose tissue inflammation and chemerin synthesis [45,46,47]. Higher chemerin mRNA expression in visceral fat in obesity was related to the degree of inflammation, thereby confirming this relationship [46].

Another study described a negative correlation of chemerin mRNA levels in subcutaneous fat with circulating chemerin protein [48]. In obese patients with non-alcoholic fatty liver disease, neither subcutaneous nor visceral fat chemerin mRNA expression was associated with its systemic levels [49]. Studies so far mostly suppose that systemic chemerin concentrations are defined by its production in adipocytes. Whether this is regulated by transcriptional and/or posttranscriptional mechanisms has not been studied in detail.

#### 2.1.2. Expression of Chemerin in Experimental Models

Whether chemerin expression is upregulated in obese adipose tissues was also analyzed in rodent models. In mesenteric fat of *Psammomys obesus*, a rodent animal model of obesity, chemerin mRNA expression was induced [25]. Female mice that were fed an atherogenic diet for eight weeks had increased chemerin protein in subcutaneous and visceral fat, and higher systemic chemerin levels than animals fed the control chow [45]. In epididymal fat of leptin deficient ob/ob and leptin receptor activity deficient db/db mice, such an induction was not observed when chemerin mRNA expression was measured [8]. In the db/db mice chemerin protein was two-fold higher when compared to the lean animals [8]. Chemerin protein was also strongly increased in the gonadal adipose tissue of male ob/ob mice. Injection of 0.5 µg leptin per gram body weight reduced chemerin protein levels in the white fat depot [50]. This indicates that leptin resistance, which accounts for metabolic disease in obese patients, may contribute to elevated chemerin protein [50,51]. In male rats that were fed a high fat diet for 12 weeks, gonadal adipose tissue chemerin protein levels were nevertheless similar to the respective controls [50].

Thus, studies in rodents mostly identified higher chemerin protein levels in obese adipose tissues. Chemerin mRNA and protein expression were analyzed in lean and obese fat and they were not concordantly regulated. Therefore, posttranscriptional processes seem to control cellular and eventually soluble chemerin protein levels.

### 2.2. Chemerin Receptor Expression in Adipose Tissues

The knock-down of chemerin or CMKLR1 in 3T3-L1 adipocyte cell line impaired adipogenesis. Further, these cells showed reduced expression of genes that are involved in glucose and lipid homeostasis [4]. Thus, chemerin–CMKLR1 signaling may enhance adipogenesis in obesity to allow for the storage of surplus lipids. However, when compared to studies measuring chemerin in obesity (see 2.1.), the expression of the respective receptors has been analyzed in less detail.

#### 2.2.1. Expression of Chemerin Receptors in Human Tissues

One study showed that CMKLR1 expression was upregulated in visceral fat of obese patients. TNF induced chemerin in human adipocytes but it had no effect on CMKLR1 mRNA [18,46]. LPS was a strong inductor of chemerin in murine 3T3-L1 adipocytes, but it did not change the CMKLR1 protein [45]. This argues against a co-regulation of chemerin and CMKLR1 expression in adipocytes by inflammatory mediators that contribute to insulin resistance and metabolic disease in obesity [52,53]. CMKLR1 is expressed by macrophages and its induction in obese fat tissues may be related to the increased number of adipose tissue resident macrophages [54].

#### 2.2.2. Expression of Chemerin Receptors in Experimental Models

An experimental model where female mice were fed an atherogenic diet for eight weeks showed higher CMKLR1 protein in subcutaneous and visceral fat [45]. In mesenteric adipose tissue of *Psammomys obesus*, CMKLR1 expression was also induced in obesity. This upregulation was, however, only seen in the fed state [25]. In epididymal adipose tissue of the ob/ob mice CMKLR1 protein levels were about five-fold higher when compared to lean wild type animals. The injection of 0.5 µg leptin per gram body weight reduced CMKLR1 protein level in this white fat depot [50]. An effect of leptin on CMKLR1 gene expression was, however, not observed in bovine adipocytes [55].

In contrast to the studies suggesting higher CMKLR1 in obese adipose tissue, further analysis in rats showed that the CMKLR1 protein was strongly reduced in the gonadal fat of male animals that were fed a high fat diet for 12 weeks [50]. CMKLR1 mRNA expression was also low in the epididymal fat of ob/ob and db/db mice [8].

Yet, whether CMKLR1 protein is indeed increased in obese fat tissue awaits clarification by additional studies. Adipocytes and stromal-vascular cells in adipose tissue express CMKLR1 [25,45], and immunohistochemical approaches have to identify the cell type specific regulation of CMKLR1 in obesity.

Adipose tissue GPR1 is primarily expressed in stromal-vascular cells [56]. Yet, the individual cells that present this receptor have not been described in detail. GPR1 mRNA was unchanged in gonadal fat of ob/ob and db/db mice when compared to lean animals [8]. GPR1 mRNA was also comparably abundant in epididymal fat of mice that were fed a control chow or a high fat diet [56]. The GPR1 protein has to be analyzed in future studies to confirm that GPR1 protein levels are indeed unchanged in obese adipose tissues.

To our knowledge, only one article regarding CCRL2 levels was published, showing that CCRL2 mRNA was significantly induced in the fat of db/db animals and tended to be higher in the leptin deficient mice [8].

### 2.3. Chemerin in the Liver

Chemerin and its receptors are expressed in the liver, but a detailed role of chemerin in hepatic function and metabolic liver diseases has not yet been explored [3]. Adeno-associated virus mediated overexpression of human chemerin in the liver of the LDL receptor knock-out mouse model led to increased systemic chemerin. This illustrates that hepatic chemerin synthesis may presumably contribute to its systemic levels. Chemerin surplus in these mice was associated with impaired insulin signaling in skeletal muscle, but not in the liver (Figure 1) [9]. Various hepatoprotective effects of recombinant mChem156 were identified in murine hepatocellular carcinoma models, but whether this isoform exists in the liver is still ambiguous [20,40].

#### 2.3.1. Expression of Chemerin in Human Tissue

Human studies that have been published so far describe normal, lower, and higher chemerin mRNA expression in the liver. Doecke et al. analyzed chemerin mRNA in controls and patients with non-alcoholic fatty liver disease and described higher hepatic levels in the overweight [57]. In morbidly obese patients, hepatic chemerin mRNA showed a trend to be induced in those probands with a BMI > 40 kg/m^2^ [58]. In a separate human study, such an association of hepatic chemerin mRNA levels with BMI was not reported [59]. Another investigation even showed reduced hepatic chemerin mRNA expression in obesity [60].

#### 2.3.2. Expression of Chemerin in Experimental Models and in vitro Systems

In the db/db mice, hepatic chemerin mRNA, but not protein, was induced [8]. Similarly, chemerin mRNA was higher in the liver of mice that were fed a high fat diet, whereas cellular chemerin protein was not concomitantly upregulated [24]. Whether the hepatic release of chemerin protein in obesity is enhanced needs to be further analyzed. Leptin deficient ob/ob mice develop severe liver steatosis, whereas the hepatic chemerin mRNA and protein were not changed [8]. Separate investigations even report reduced hepatic chemerin levels in murine obesity [60].

Inflammatory cytokines, like TNF and LPS, are elevated in obesity and upregulated adipocyte chemerin [18,45,47]. These factors did not change chemerin expression in hepatocytes [24,47,57]. Leptin did not induce human hepatocyte chemerin levels [24].

Current data show that obesity is not necessarily associated with altered hepatic chemerin levels. For which reasons the different research groups identified normal, higher, and lower chemerin expression in the liver of overweight humans and mice is still an open question.

### 2.4. Chemerin Receptors in the Liver

Analysis of CMKLR1, CCRL2, and GPR1 in the liver of ob/ob and db/db mice revealed that only GPR1 mRNA was strongly decreased [8]. In mice that were fed a high fat diet hepatic CMKLR1 mRNA was found to be reduced [61]. Human studies described a positive or no association of hepatic CMKLR1 levels with BMI [57,62]. The CCRL2 mRNA levels were not different in normal-weight and overweight patients [63].

While CMKLR1 mRNA was found to be expressed in various liver cells, including hepatocytes, hepatic stellate cells, endothelial cells, and Kupffer cells, CCRL2 seems not to be expressed in hepatocytes [61,63]. The downregulation of CMKLR1 in Kupffer cells by a phosphatidyl inositol 3-kinase inhibitor improved hepatic insulin resistance and inflammation, suggesting that the chemerin–CMKLR1 signaling pathway contributes to metabolic disease in obesity by modulating the function of these immune cells [64]. Here, it has to be considered that resolvin E1 is a further ligand of CMKLR1, which is well described to have a function in the resolution of inflammation [65].

### 2.5. Chemerin and its Receptors in Skeletal Muscle

Analysis of chemerin, CMKLR1, CCRL2, and GPR1 in skeletal muscle of ob/ob and db/db mice revealed that CMKLR1 mRNA was strongly increased in both of the strains. Chemerin was only upregulated in the muscle of db/db mice [8]. GPR1 mRNA was markedly reduced in soleus muscle but not in the gastrocnemius muscle of mice that were fed a high fat diet [56].

Although chemerin induces skeletal muscle insulin resistance [18] (Figure 1), the expression of the corresponding receptors in muscle was not studied in detail. Analysis of chemerin and its receptors in skeletal muscle tissues of rodents and humans is needed to further understand the role of this adipokine in skeletal muscle insulin resistance.

## 3. Chemerin Isoforms and Activity in Adipose Tissue and Serum

### 3.1. C-terminal Processing and Activity of Chemerin Isoforms

The open reading frame of the human chemerin gene codes for a 163 amino acid protein. The secreted form is shorter by 20 amino acids due to the removal of the N-terminal signal peptide [2]. N-terminally cleaved chemerin with an intact C-terminus is traditionally named hChem163, although it only consists of 143 amino acids [19]. The number in the abbreviation thus corresponds to the respective amino acid position of the full-length protein. The analogous murine chemerin protein is one amino acid shorter and it is designated as mChem162. Human and murine chemerins are 64% identical with a similarity of 78% [6,66].

HChem163 and mChem162 need to be processed at their C-termini to become active proteins. Proteolytic cleavage by extracellular proteases at distinct C-terminal sites generates highly active, high active, moderate active, and inactive isoforms [2,66] (Figure 2). The similarity between human and murine chemerin suggests that protease cleavage sites are conserved between these two species [66]. Chemerin isoform bioactivity was mostly determined by chemotaxis assays and/or the analysis of intracellular calcium release. These assays revealed that hChem157 had the highest activity [2,13,67]. Analysis was undertaken with primary monocyte-derived dendritic cells and murine pre-B lymphoma L1.2 cells overexpressing human CMKLR1 [13,67].

HChem157 is produced by cathepsin K and L, and human leukocyte elastase cleavage of hChem163 [67,68] (Figure 2). HChem156 was as active as hChem157 when it was analyzed in a signal transduction assay using cells with CMKLR1 overexpression [69]. The analysis of Ca^2+^ mobilization using L1.2 cells stably expressing human CMKLR1 revealed that the shorter isoform was by far less active than hChem157 [70]. HChem156 originates from hChem163 by cathepsin G and chymase cleavage [70,71] (Figure 2). The serine protease kallikrein 7 also produced hChem156 from prochemerin, and it may contribute to chemerin activation in the skin [69]. HChem158 and hChem155 are low active isoforms. HChem158 was produced by plasmin, tryptase, and factor XIa cleavage [67,72] (Figure 2). HChem155 derives from hChem163 by elastase, proteinase 3, or tryptase mediated proteolysis [67,71]. Elastase, cathepsin K and L, and chymase also contribute to chemerin inactivation by further C-terminal processing [67,68,71] (Figure 2).

Angiotensin converting enzyme is a carboxypeptidase and it removes C-terminal dipeptidyl residues amongst others from angiotensin I to obtain the vasoconstrictor angiotensin II [73], thereby controlling blood pressure. This enzyme was shown to produce hChem152 from hChem154 [74]. Chemerin affects several pathways that control blood pressure, indicating a function of this adipokine herein [17,75] (Figure 1). Of note, systemic chemerin was increased in patients with hypertension indicating that this adipokines may increase blood pressure [76,77]. Surprisingly, hypertensive male rats with knock-out of chemerin had elevated mean and systolic blood pressure. In female rats, chemerin deficiency was associated with lower pressures, as one would have expected [78]. A second study using male rats observed decreased arterial blood pressure upon chemerin knock-down [79]. These studies show that chemerin has a causative function in blood pressure control, which is modified by gender. Analysis of chemerin isoform distribution in females and males and normo- and hypertensive patients may clarify the role of angiotensin converting enzyme mediated proteolysis of chemerin herein. Murine mChem156, which is a highly active isoform, increased blood pressure in male mice [75]. Whether this isoform is present in hypertension in biologically relevant concentrations needs to be elucidated.

Proteolytic cleavage sites for human chemerin, which are described above, are conserved in mice [66]. Here, mChem156 and mChem155 had nearly similar activities in Ca^2+^ mobilization and chemotaxis assays. A 15- to 20-fold lower potency was measured for mChem161 and mChem157 [66]. The authors of this paper failed to express recombinant mChem162, and mChem161 was detectable in the supernatants of the transfected cells. It was suggested that mChem161 had similar properties to mChem162. MChem154 did neither have chemoattractant properties nor did it induce Ca^2+^ release [66].

Activities of the various chemerin isoforms may deviate from the activity ranking that is described above in men and mice when biologic effects other than chemotaxis and Ca^2+^ influx are tested. For analysis of chemerin isoform activity, mostly cells with CMKLR1 overexpression were used [66,71]. These results may differ from chemerin induced GPR1 signaling that has been studied in less detail. Which of the receptors is more relevant for the biological and pathophysiological effects of chemerin is mostly unknown.

Processing of chemerin by various proteases produces different isoforms that differ in their activity. Short variants are mostly inactive and they may even antagonize the active isoforms [80]. This demonstrates that a complex regulatory network controls chemerin bioactivity. The analysis of total chemerin protein levels cannot provide appropriate information regarding its biologic activity.

There was even discrepancy when measuring total chemerin protein with a pan-chemerin ELISA or isoform specific ELISAs. The EC_50_ values (the concentration of antibodies showing half-maximal binding) of the human pan-chemerin ELISA were lower for hChem163, hChem158, and hChem155 when compared to hChem157. Thus, the analysis of samples with a low level of Chem157 using this pan-chemerin assay will underestimate the actual chemerin concentration [81]. EC_50_ values for the different murine chemerin isoforms vary by up to two-fold when using a commercial pan-chemerin ELISA kit. Using this assay to measure total chemerin will give too high values in samples that contain mChem155 and mChem154, and too low values when mChem162 and mChem157 are the abundant variants [66].

Chemerin isoforms in body fluids, cell supernatants, and tissues can be exactly defined by liquid chromatography/mass spectroscopy [81]. Isoform specific chemerin ELISAs have been established and they were used for the quantification of the individual human and murine variants [66,81]. The so-called Tango bioassay was developed to determine G protein-coupled receptor ligand induced beta-arrestin 2 recruitment [82] and it was applied to measure chemerin bioactivity in cell culture supernatants and plasma [47,83,84].

### 3.2. Quantification of Chemerin Activity with the Tango Bioassay

The Tango bioassay can quantitatively measure CMKLR1 and GPR1 beta-arrestin 2 activation [47]. In this analysis, human embryonic kidney (HEK) 293T cells that constitutively express a tobacco etch virus (TEV) protease fused to beta-arrestin 2 are used. Furthermore, these cells encode a reporter gene whose expression is induced by a transcriptional-transactivator (tTA). To test for the bioactivity of chemerin, the HEK293T cells are transfected with a plasmid encoding either CMKLR1 or GPR1. The C-terminus of these receptors carries a TEV N1a protease cleavage site and the tTA. Chemerin induces the recruitment of the beta-arrestin 2–protease fusion protein to GPR1-tTA or CMKLR1-tTA. The protease cleaves the TEV N1a site and it releases tTA, which passes into the nucleus. Here, tTA enhances the transcription of the respective reporter gene. By using appropriate standards, the biological activity of chemerin in different samples can be quantitatively determined. One has to keep in mind that only the beta-arrestin 2 dependent pathways are analyzed by these activity assays.

The human CMKLR1 based Tango bioassay revealed that murine chemerin activated this receptor. CMKLR1 activation was confirmed in murine adipocyte and hepatocyte conditioned media, demonstrating that both cell types produce bioactive chemerin [47]. TNF induced total chemerin protein in 3T3-L1 adipocytes and chemerin bioactivity in the supernatants was accordingly increased [47]. Mice that were injected with TNF had higher serum chemerin, which had a higher potency to activate CMKLR1 [47]. Therefore, the inflammatory cytokine TNF enhances chemerin production and concentrations of bioactive chemerin isoforms.

TNF is increased in obesity and it contributes to insulin resistance [85]. One may assume that low grade chronic inflammation in obesity may activate chemerin, which subsequently impairs skeletal muscle insulin resistance. However, recent research that is summarized in the next paragraphs did not report on higher chemerin activity in obesity.

### 3.3. Ex-vivo Analyzed Systemic Chemerin Activity in Human and Murine Obesity

Among the most widely used rodent models in obesity and type 2 diabetes research are the ob/ob mice and the db/db mice [86]. In both strains, the total serum chemerin protein and ex-vivo activation of CMKLR1 were about two-fold higher than in C57BL/6 controls [8]. A second study did not find higher CMKLR1 activity in the ob/ob mice [83]. The ob/ob mice in both of the studies were about three months old [8,83], excluding that different age contributed to this discordant findings. GPR1 activation was measured in the second study and it was found to rise in parallel to the total chemerin protein [83]. In mice that were fed a high fat diet for 14 weeks, CMKLR1 and GPR1 activities were induced. The ratio of bioactive chemerin to total chemerin was higher in the overweight animals with respect to CMKLR1 activation. This effect was not identified with regard to GPR1 activity [83]. This indicates that these receptors may vary in their affinity for distinct chemerin isoforms.

These experiments clearly show that chemerin bioactivity does not always correspond to total systemic chemerin protein levels, at least when measuring chemerin-induced beta-arrestin 2 recruitment [8,83].

To our knowledge, CMKLR1 activation was measured in only one study in human obesity. Here, the total circulating chemerin protein was detected by a pan-chemerin ELISA and ex-vivo CMKLR1 activity was determined with the beta-arrestin 2 Tango bioassay [84]. Obese females had higher total serum chemerin and unchanged CMKLR1 activation when compared to normal-weight women. Thus, the ratio of bioactive to total chemerin levels was significantly reduced in the obese [84]. This even reflects impaired chemerin activity, despite higher total systemic chemerin protein.

In this study, circulating chemerin protein and CMKLR1 activation were also investigated in the postprandial phase. Chemerin levels and ex-vivo CMKLR1 activation were similar in the fasted and the postprandial state [84]. This is in line with a previous analysis in a cohort of healthy individuals, where serum chemerin was similar in the fasted state, 1 h and 2 h after oral glucose uptake [87]. Impaired postprandial glucose and lipid clearance contribute to metabolic disease in obesity [88]. Current data principally argue against a function of chemerin herein.

### 3.4. Chemerin Isoforms in Serum

In human plasma, inactive, full-length chemerin (hChem163) was by far the most abundant form [81]. HChem157 was about 25-fold and hChem155 about 300-fold less present (Figure 3). The plasma levels of hChem157 were below the EC_50_ of about 1.2 +/- 0.7 nm, which was determined in the calcium mobilization assay using CMKLR1 expressing cells [80,81]. Type of diet, gender, and age were not associated with changes in chemerin isoform distribution, which was relatively invariable in lean volunteers during three years of follow up [81]. Systemic chemerin may thus reflect a reservoir of biologically inactive prochemerin that can be very quickly activated upon request.

Total chemerin protein that was analyzed by an ELISA supposed to measure all isoforms was higher in the overweight and the obese. The concentration of the chemerin isoforms hChem163, hChem157, and hChem155 was, however, not markedly changed in the overweight individuals or in obesity (Figure 3). So-called unattributed chemerin variants appeared in the plasma of the overweight and the obese. These chemerin forms were detected by commercially available chemerin ELISAs, but not by ELISAs that were designed to specifically measure hChem163, hChem157, or hChem155. These short variants had relatively large C-terminal truncations and hChem144 was significantly induced in the plasma of the obese [81]. The truncated chemerin variants were most likely biologically inactive, demonstrating that higher chemerin protein in obesity was not linked to increased chemerin bioactivity. Positive correlation of unattributed, truncated chemerin levels and elastase in blood suggests that this enzyme may participate in chemerin processing and inactivation. Because truncated forms of chemerin were not present in adipose tissues, proteolysis has to occur in the extracellular space and/or in blood [81].

In mice that were fed a high fat diet for 26 weeks (starting with six week old animals), the total plasma chemerin was increased and this was attributed to higher levels of inactive mChem162. Of note, 12 weeks high fat diet feeding raised mChem157 and mChem156 in plasma, and here, these two isoforms represented about 55% of systemic chemerin. Indeed, the levels of mChem157 and mChem156 were highest in the 12 week high fat diet fed mice when compared to animals that were fed a low fat diet for 12 or 26 weeks and the mice fed a high fat diet for 26 weeks. The authors of this work suggested that production of the highly active chemerin form coincides with the time of early adipogenesis that is promoted by chemerin [4,66]. The biologically modest-active mChem157 may be the source for mChem156 production, a hypothesis that has to be proven in future studies [66]. The levels of the isoforms mChem157 and mChem156 were similar in mice that were fed a low or a high fat diet for 26 weeks. MChem155 and mChem154 isoforms were not significantly different in the lean and the obese mice and when analyzed 12 or 26 weeks after starting the experiments [66]. ELISA only detected mChem154 at low levels in the older animals, while in the younger mice this isoform was not measured at all. MChem155, which was as active as mChem156, accounted for about 6.4–7.7% of total plasma chemerin, irrespective of age and diet [66].

The total chemerin protein levels and mChem162 were higher in the older mice, irrespective of the diet, while further isoforms were unchanged, showing that the aging of mice is associated with elevated levels of biologically inactive plasma chemerin [66].

### 3.5. Chemerin Isoforms in Adipose Tissues

Serum chemerin is supposed to be mainly derived from adipose tissues [1,2,8,47]. Unexpectedly, an analysis of chemerin isoforms in human subcutaneous and omental fat revealed that hChem163, which is the most prominent chemerin protein in the circulation, was not the dominant form in these tissues [81]. Indeed, in human subcutaneous adipose tissue, hChem157 represented about 90% of the three isoforms analyzed, which were hChem155, hChem157, and hChem163. In omental fat, about 80% of the chemerin isoforms was hChem155, while about 20% was hChem157 (Figure 3). HChem163 constituted approximately 1–2% of total omental fat chemerin protein [81] (Figure 3). The levels of the active hChem157 isoform in the fat depots were 1,000-fold higher than in plasma and by far sufficient to induce calcium mobilization in CMKLR1 expressing cells [80,81]. The total chemerin protein was more abundant in the omental fat depot, but it was unchanged in both adipose tissue depots in obesity. Truncated chemerin isoforms that were detected in plasma of the obese did not exist in the fat tissues [81] (Figure 3).

In murine epididymal adipose tissue of 18 week old mice, mChem155 was the predominant form, representing nearly 100% of total chemerin protein. This was completely changed when analyzing this fat depot 14 weeks later. Here, 85–91% was mChem162, approximately 5 to 10% was mChem154, while mChem155 was about 1–2%. MChem157 and mChem156 were not detected in the adipose tissues of the mice. High fat diet feeding for 12 or 26 weeks did neither affect total chemerin protein nor chemerin isoform distribution [66]. In this animal model, mChem155, but not mChem156, was the active chemerin isoform in the intraabdominal fat depot [66]. Chymase positive mast cells are localized in adipose tissues and this protease may generate mChem155 from prochemerin [70,89].

In the older mice, total chemerin protein in epididymal fat even declined in obesity, and it was negatively correlated with body weight, amount of fat, and plasma total chemerin protein [66]. Adipocytes differentiated from murine primary mesenchymal stem cells secreted chemerin and mChem152, -156, -158, -159, -160, -161, and -162 were detected in the supernatants [22]. Chemerin isoform abundance was, therefore, completely different in murine epididymal fat tissue and in-vitro differentiated murine adipocytes. This indicates that proteases that were produced by stromal-vascular cells contribute to chemerin processing in the fat depots.

Brown adipose tissue expressed about 15-fold lower levels of total chemerin protein when compared to epididymal fat of mice. While high fat diet feeding for 12 weeks did not change chemerin protein, it was induced in obesity when this diet was given 14 weeks longer [66].

Chemerin isoform distribution in adipose tissue was completely different from the variants that were found in the circulation. Prochemerin was released into the circulation, while further processed chemerin was present in fat tissues. Isoform distribution was not changed in obesity in mice and men. In murine fat tissue chemerin isoforms abundance was mostly affected by age and this may be related to the phase of life with increased adipogenesis [66,81].

## 4. Conclusions

Chemerin’s importance in physiological and pathophysiological processes has been illustrated in numerous clinical and experimental studies. The role of chemerin is, however, incompletely understood. Systemic chemerin is increased in obese mice and humans, while its bioactivity is not concordantly changed. The signal transduction pathways and the physiologic function of GPR1 are mostly unstudied. The biologic activity of the chemerin isoforms binding to this receptor has not been characterized in detail. Chemerin processing seems to be changed in obesity. The respective proteases that are involved herein are still not defined. Synthetic chemerin-derived peptides that resemble the C-terminal amino acids of this adipokine reduce inflammation and enhance phagocytosis [41,90], and they may be used as therapeutic agents for the treatment of metabolic diseases and possibly further chronic inflammatory disorders.

## Figures and Tables

**Figure 1 ijms-20-01128-f001:**
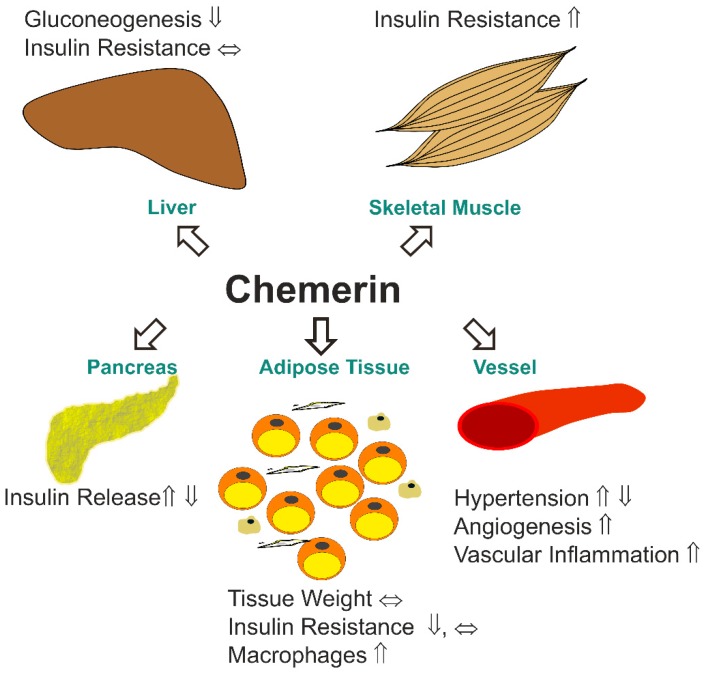
Effect of chemerin on the metabolic status of different organs (inconclusive results indicated by reverse arrows). Data published so far mostly agree that chemerin impairs skeletal muscle insulin response. This was not observed in the liver, here gluconeogenesis was enhanced in chemerin deficient mice. The function of chemerin on blood pressure was modified by gender. Chemerin further stimulated angiogenesis and vascular inflammation. Adipose tissue weight was not changed by chemerin. This adipokine may even improve insulin response of fat tissue although the number of adipose tissue resident macrophages was increased. Stimulatory and inhibitory effects of chemerin on glucose-induced release of insulin by pancreatic beta-cells was reported. Inconclusive findings may be partly explained by the different models studied.

**Figure 2 ijms-20-01128-f002:**
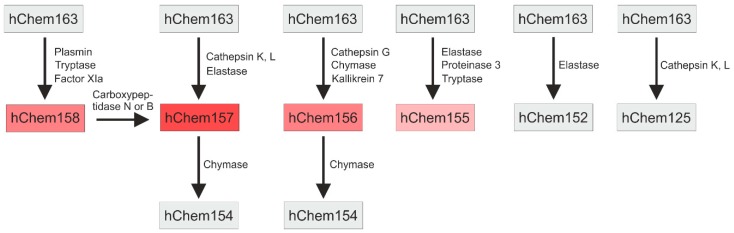
Processing of human chemerin. The proteases contributing to C-terminal processing of chemerin and the respective isoforms generated are shown. Inactive isoforms are in grey boxes, biologic active isoforms in red boxes. The intensity of the red color corresponds to the activity of the chemerin isoform (intense red: very active isoform). Activity has been mostly analyzed using Ca^2+^ flux and migration assays in chemokine-like receptor 1 (CMKLR1) expressing cells. Angiotensin converting enzyme converts hChem154 to hChem152. This is not shown in the figure. HChem154 is produced by different proteases and the enzymes upstream of angiotensin converting enzyme have not been identified yet.

**Figure 3 ijms-20-01128-f003:**
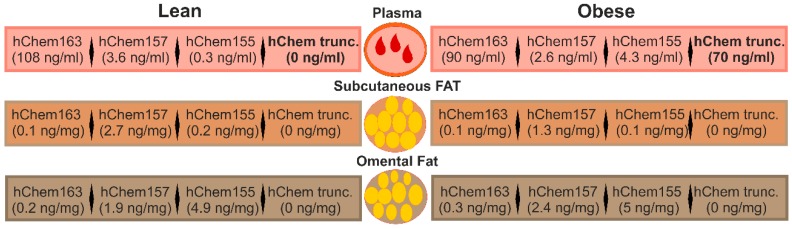
Chemerin isoform distribution in human obesity. The different chemerin isoforms identified in human plasma, subcutaneous and omental adipose tissues are shown. The concentrations analyzed by chemerin isoform-specific ELISAs in the tissues are given as ng chemerin/mg adipose tissue. Truncated (trunc.) isoforms are relatively short and they are most likely not biologically active. Comparison of the chemerin levels in the lean and the obese probands revealed that only the truncated forms in plasma are significantly induced in the latter.

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
