# Peer review of "Chemerin Isoforms and Activity in Obesity"

_ijms, 2019, doi:10.3390/ijms20051128_

Reviewer 1 Report

Summary / significance:

This review by Buechler et al. focuses on current articles highlighting recent research contributing to the understanding of chemerin expression and action in various disease settings. The authors give an introductory overview of chemerin expression and according chemerin receptors. They particularly point out the diverse abundance of chemerin isoforms and conclude that these subtle differences are essential understand its mode of action. Function of the adipokine/chemokine chemerin (RARRES2 gene) and its main receptor (CMKLR1/ChemR23) is matter of research in diseases accompanied by unwanted adipogenesis such as seen in obesity, but also in chronic inflammatory diseases. Chemerin levels were found increased in individuals with high BMI, blood pressure and insulin resistance, indicating an involvement in metabolic alterations. Elevated chemerin levels were also traced in type 2 diabetic patients and chronic kidney disease. On the other hand, CMKLR1 signalling regulates inflammatory pathways, and seems to be involved in insulin resistance. The authors specifically highlight the importance of post-translational isoform abundance that seems to essentially determine its efficacy. Hence, to determine the actual chemerin activity in biologic fluids, a quantitative in vitro assay (the Tango assay) that determines GPCR-activation as read-out is recommended.

Level of interest/merit:

Understanding of chemerin isoform abundance and function is of high interest to the scientific readership to develop novel powerful tools for use in anti- and/or pro-adipogenic or inflammatory therapy. This is an interesting review of IJMS focussing on studies that investigate chemerin abundance and function in general physiological and clinical contexts.

There are some comments:

This article is written clearly, the contents are accurately and coherent, and the English level is good. Several points could be addressed to make the text more exciting and fluent to read:

• The introductory part could be improved by including a graph depicting the chemerin signaling path.

• For clearness, the contents of each paragraph could be organized in the order of 1) expression in human tissue, followed by 2) expression in experimental rodent models and 3) in vitro systems.

• Maybe split section 3.1. into two sections, one describing the processing, and the next describing

according measurement methods, including the Tango assay?

Detailed comments:

Generally, please stick to past tense when describing findings from the reviewed studies, and check English phrasing and spelling to improve legibility of the text.

Line 30: maybe better: “Chemerin is also an adipokine that regulates….”

Line 42: “As an adipokine, chemerin is released by adipocytes, but also hepatocytes produce…”

Line 45: “Hence, association of systemic chemerin levels with metabolic syndrome…”

Line 48: include gene ID: RARRES2

Line 68: “This review article briefly describes the various C-terminal processing forms of chemerin. Expression of chemerin and its receptors….”

Line 73: “Chemerin is expressed most abundantly in white adipose tissue, in the liver and to lesser extent in brown adipose tissue…”

Line 91f: “…was analyzed in rodent animal models.”

Line 111f: “Knock-down of chemerin…in 3T3-L1 adipocyte cell line impaired adipogenesis. Further, these cells showed reduced expression of genes involved in glucose and lipid homeostasis…”

Line 114f: “..expression of the respective receptors has been analyzed in less detail. One study showed that CMKLR1 expression is upregulated in visceral fat of human obese patients….”

Line 150f: “Adeno-virus mediated overexpression of human chemerin….”

Author Response

We are very grateful to the reviewer for fair and very helpful comments.

This article is written clearly, the contents are accurately and coherent, and the English level is good. Several points could be addressed to make the text more exciting and fluent to read:

Thank you for this kind comment.

• The introductory part could be improved by including a graph depicting the chemerin signaling path.

We added a short paragraph on the signaling molecules involved in the Introduction.

• For clearness, the contents of each paragraph could be organized in the order of 1) expression in human tissue, followed by 2) expression in experimental rodent models and 3) in vitro systems.

Paragraphs in chapter 2 were reorganized as suggested (human tissue versus rodent models), namely 2.x.1, 2.x.2 with x being chemerin in adipose tissue, or chemerin receptor, or chemerin in liver, or chemerin receptor in liver, or chemerin and receptors in skeletal muscle.  There are not sufficient data to include “in vitro systems”.

• Maybe split section 3.1. into two sections, one describing the processing, and the next describing according measurement methods, including the Tango assay?

We would prefer the present sections because splitting would mandate some redundancy in the following paragraph.

Detailed comments:

Generally, please stick to past tense when describing findings from the reviewed studies, and check English phrasing and spelling to improve legibility of the text.

Line 30: maybe better: “Chemerin is also an adipokine that regulates….”

Line 42: “As an adipokine, chemerin is released by adipocytes, but also hepatocytes produce…”

Line 45: “Hence, association of systemic chemerin levels with metabolic syndrome…”

Line 48: include gene ID: RARRES2

Line 68: “This review article briefly describes the various C-terminal processing forms of chemerin. Expression of chemerin and its receptors….”

Line 73: “Chemerin is expressed most abundantly in white adipose tissue, in the liver and to lesser extent in brown adipose tissue…”

Line 91f: “…was analyzed in rodent animal models.”

Line 111f: “Knock-down of chemerin…in 3T3-L1 adipocyte cell line impaired adipogenesis. Further, these cells showed reduced expression of genes involved in glucose and lipid homeostasis…”

Line 114f: “..expression of the respective receptors has been analyzed in less detail. One study showed that CMKLR1 expression is upregulated in visceral fat of human obese patients….”

Line 150f: “Adeno-virus mediated overexpression of human chemerin….

Thank you for your very detailed suggestions. The text was corrected according to your suggestions. We made the following changes: 

“Line 114f: “..expression of the respective receptors has been analyzed in less detail. One study showed that CMKLR1 expression is upregulated in visceral fat of human obese patients….” Here “human” was left out. This is already clear by talking about patients.

“Line 150f: “Adeno-virus mediated overexpression of human chemerin…” This was not changed because adeno-associated virus differs from adenovirus. The first may be used for human therapy, is less immunogenic and expresses the transgene up to 6 months.

Reviewer 2 Report

This review reports a very detailed description of the various isoforms of chemerin vs. chemerin receptors and of their putative role in obesity. Extensive information on the tissue localization is provided, and details on the various isoforms known to be involved directly or indirectly in obesity states are given.

Major points

None

Minor points

What  is most probably missing in this review is a synthetic description of the physiopathological context in which chemerin operate. A summary picture on the various signal pathways activated/inhibited by the ligand/receptor interaction might help allocating all the information given.

Author Response

We want to thank the reviewer for his advice, Adding data on the role of chemerin really improved the article.

Minor points

What  is most probably missing in this review is a synthetic description of the physiopathological context in which chemerin operate. A summary picture on the various signal pathways activated/inhibited by the ligand/receptor interaction might help allocating all the information given.

 We partly addressed this important advice. We included figure 1 showing the effects of chemerin on organs affected in obesity.

We also added a paragraph summarizing the signal pathway activated by chemerin.

 Data related to all of these findings are mostly not concordant. E.g. it is not clear whether chemerin blocks or stimulates insulin release from pancreatic beta-cells. Chemerin may further activate or block NFkappaB activity. Thus signaling is very complex and has to be carefully evaluated by comparing culture conditions, time of incubation, concentrations of chemerin and so on. This might be an issue for a separate review article.